# CD44 as a novel therapeutic target in pulmonary arterial hypertension: Insights from multi-omics integration and molecular docking

**Wei Chen[1], Lingling Zhang[2], Haiyan Qi[3,4]***

1 School of Economics & Management, Jiangxi Agricultural University, Nanchang, Jiangxi, China,
2 Department of Pathology, Nanchang People's Hospital Affiliated to Nanchang Medical College, Nanchang, Jiangxi, China, 3 School of Computer Science & Engineering, Jiangxi Agricultural University, Nanchang, Jiangxi, China, 4 School of Information Management and Mathematics, Jiangxi University of Finance and Economics, Nanchang, Jiangxi, China

* qihaiyanwanna@163.com

## Abstract

Pulmonary arterial hypertension (PAH) is a progressive and often fatal disorder characterized by increased pulmonary vascular resistance and subsequent right heart failure. Inflammation plays a pivotal role in the pathogenesis of PAH, and recent studies have highlighted the potential therapeutic significance of targeting inflammatory pathways. This study investigates the role of CD44, a cell surface receptor, in the inflammatory processes underlying PAH. By analyzing bulk RNA-seq data from idiopathic pulmonary hypertension (IPAH) patients and conducting single-cell RNA-seq analysis on pulmonary arterial cells, we identified CD44 as a key modulator of inflammation. Our findings suggest that elevated CD44 expression is not only in T cells but also prominently in pulmonary artery smooth muscle cells (SMCs), suggesting its involvement in vascular inflammation and remodeling. Molecular docking studies revealed a potential interaction between CD44 and progesterone, an anti-inflammatory drug and immunomodulator, and this indicates a novel avenue for therapeutic intervention. The results support the hypothesis that targeting CD44 may reduce inflammation and improve clinical outcomes in PAH patients.

## 1. Introduction

Pulmonary arterial hypertension (PAH) is a progressive and lethal disorder defined by a mean pulmonary artery pressure exceeding 20 mmHg at rest, accompanied by pathological vascular remodeling, endothelial dysfunction, and perivascular inflammation [1,2]. The disease culminates in right heart failure, with a median survival of 5–7 years post-diagnosis despite modern therapies [2]. Current treatments, including phosphodiesterase-5 inhibitors, endothelin receptor antagonists, and prostacyclin analogs, primarily target vasodilation and hemodynamic stabilization [2]. However,

**Data availability statement:** All RNA-seq data are available from the NCBI Gene Expression Omnibus (GEO) database (accession number GSE144274 and GSE228644).

**Funding:** This work was supported by grants from Scientific Research Foundation of the Education Department of Jiangxi Province, China (No. GJJ2200446 to H.Q.) and Jiangxi Provincial Graduate Innovation Special Foundation (No. YC2024-B178 to H.Q. and No. YC2023-B166 to W.C.).

**Competing interests:** The authors have declared that no competing interests exist.

these approaches fail to address the underlying inflammatory and proliferative mechanisms driving vascular pathology [3,4], which underscore the urgent need for novel therapeutic strategies.

Emerging evidence highlights that inflammation is increasingly recognized as a hallmark of PAH, with infiltrating immune cells (e.g., T lymphocytes, macrophages) secreting pro-inflammatory cytokines such as TNF-α, IL-6, and IL-1β [5,6]. These cytokines activate NF-κB signaling, promoting SMC proliferation, endothelial apoptosis, and extracellular matrix deposition [7,8]. Lung endothelial dysfunction triggers immune cell adhesion and inflammatory mediator release, driving perivascular inflammation, which in turn exacerbates vascular remodeling and PH progression [9]. Recent studies suggest that anti-inflammatory therapies, including corticosteroids and non-steroidal anti-inflammatory drugs, attenuate vascular remodeling in preclinical models, yet their clinical translation is hindered by systemic side effects and incomplete mechanistic understanding [10–12].

CD44, a transmembrane glycoprotein, mediates leukocyte adhesion, migration, and activation through interactions with hyaluronic acid (HA) and other extracellular matrix components [13]. In cancer and autoimmune diseases, CD44 facilitates inflammatory cell recruitment and cytokine release, making it a potential target for immunomodulation [14,15]. Recent studies suggest that CD44 may play a role in PAH by modulating immune cell activation and vascular remodeling [16]. Proinflammatory IL-1β upregulates CD44, driving phenotypic modulation and contributing to arteriosclerosis pathogenesis [17]. In murine and human models, CD44 drives SMC proliferation, dedifferentiation, and activation of VCAM-1 [18]. However, the precise mechanisms by which CD44 contributes to PAH pathogenesis remain poorly understood.

This study employs a multi-omics approach to unravel CD44's contribution to PAH pathogenesis. By integrating bulk and single-cell transcriptomics and molecular docking, we reveal CD44's involvement in inflammatory processes and identify progesterone as a potential modulator of CD44 activity. Our findings provide a mechanistic foundation for CD44-targeted therapies, bridging a critical gap in PAH treatment.

## 2. Materials and methods

### 2.1. Transcriptomic analysis of bulk RNA-seq data

Bulk RNA sequencing data from IPAH patients (n = 4) and healthy controls (n = 4) were obtained from the Gene Expression Omnibus (GEO) database (GSE144274) [19], with data access and download completed on August 12, 2024. Differential expression analysis to identify differentially expressed genes (DEGs) was performed using the limma package (v3.54.0) in R (v4.3.2). A design matrix was constructed for linear modeling, where disease status (IPAH vs. Control) was specified as the primary variable of interest, and age and sex were included as covariates to account for potential confounding effects. Genes with |log2 fold change| > 1 and adjusted p-value < 0.05 (Benjamini-Hochberg correction) were classified as significant. A total of 1,651 DEGs were identified, including 670 upregulated and 581 downregulated genes. The full list is provided in S1 Table.

## 2.2. Functional annotation: GO and KEGG pathway enrichment analysis

GO and KEGG pathway analyses were performed using the clusterProfiler R package (v4.6.2). Enrichment significance was assessed via hypergeometric testing, with a p-value cutoff of 0.05 and a q-value cutoff of 0.2 to define statistically significant terms. Dot plots were generated to visualize enriched terms, with gene counts reflected by point size, and statistical significance (adjusted p-value (Benjamini-Hochberg correction)) indicated by color intensity. The complete enrichment results, including term descriptions, gene counts, and FDR values, are provided in S2 Table.

## 2.3. Integration of inflammation-related genes from the GeneCards database

To prioritize inflammation-associated DEGs, the 670 upregulated genes were intersected with the GeneCards database (www.genecards.org) using the keyword "inflammation" and a relevance score > 5. This yielded 47 overlapping genes, including CD44.

## 2.4. Protein-protein interaction (PPI) network construction and hub gene identification

The 47 inflammation-related genes were submitted to the STRING database (v11.5) to construct a PPI network, with interaction confidence set to > 0.7 and disconnected nodes hidden. The network was imported into Cytoscape (v3.9.1) for visualization and topological analysis. Hub genes were identified using the cytoHubba plugin in Cytoscape, based on multiple topological analysis algorithms, including Betweenness, Bottleneck, Closeness, EPC (Edge Percolated Component), Radiality, Degree, MNC (Maximum Neighborhood Component), EcCentricity, and Stress. CD44 emerged as the top hub gene in the top 10 nodes ranked by these methods.

## 2.5. Single-cell RNA sequencing data processing and clustering

The single-cell RNA-seq dataset GSE228644 [20], comprising pulmonary artery cells from 3 IPAH patients and 3 controls, was processed using Seurat (v4.3.0), with data access and download completed on August 14, 2024. Low-quality cells (mitochondrial gene percentage > 20%, detected genes < 200) were excluded. Data normalization was performed via SCTransform, and variable features were selected (2,000 genes). Principal component analysis (PCA) identified 30 significant PCs, and UMAP dimensionality reduction was applied for clustering (resolution = 0.8). Cell types were annotated manually using marker genes in each cluster. Lung-derived scRNA-seq clusters were annotated into major cell types including fibroblasts/myofibroblasts (clusters 0, 10, 15, 23), T cells (clusters 1, 2, 5), NK cells (cluster 11), macrophages/monocytes (clusters 4, 6, 9, 14, 20, 22), endothelial cells (clusters 7, 12), epithelial cells (clusters 16, 18, 21), and pericytes (cluster 19). Marker gene statistics and cell type annotations for all clusters are provided in S3 Table. Differential expression of CD44 across clusters was assessed using the Wilcoxon rank-sum test.

## 2.6. Molecular docking and binding affinity validation

The crystal structure of the CD44 HA-binding domain (PDB ID: 1UUH) was retrieved from the RCSB Protein Data Bank. Progesterone (CID: 5994) was prepared using AutoDockTools (v1.5.7) by adding Gasteiger charges and merging non-polar hydrogens. Docking simulations were performed in AutoDock Vina (v1.2.3) with a grid box centered on the HA-binding site. PyMOL (v2.5.2) was used for interaction visualization.

## 2.7. Statistical analysis

Data are presented as mean ± SEM. Statistical comparisons were made using the Wilcoxon rank-sum test. $p < 0.05$ was considered significant.

## 3. Results

### 3.1. Differential gene expression analysis reveals inflammatory dysregulation in IPAH patients

Bulk RNA sequencing analysis of human samples from 4 IPAH patients and 4 healthy individuals (GSE144274) identified 670 significantly upregulated genes and 581 downregulated genes (|log2 fold change|>1, adjusted p-value<0.05) (Figs 1A and S1). Additionally, CD44 expression was significantly higher in the IPAH group (Fig 1B). To explore the potential biological functions of DEGs, GO and KEGG enrichment analyses were conducted separately for the upregulated and downregulated gene sets. GO enrichment analysis revealed that upregulated genes were predominantly associated with cell division–related biological processes, including chromosome segregation, nuclear division, mitotic cell cycle phase transition, and DNA replication (Fig 1C). These findings suggest increased proliferative activity in the IPAH group. Conversely, downregulated genes were enriched in processes related to response to oxygen levels, response to hypoxia, extracellular matrix organization, and response to endoplasmic reticulum stress, highlighting impaired cellular stress responses and tissue remodeling in IPAH (Fig 1D). KEGG pathway analysis further supported these findings, showing enrichment in pathways such as cell cycle, DNA replication, oocyte meiosis, base excision repair, and progesterone-mediated oocyte maturation (Fig 1E). Additionally, the downregulation of TNF signaling, chemokine activity and inflammatory pathways, including "cytokine-cytokine receptor interaction" and "NF-κB signaling", suggests a state of immune dysregulation or exhaustion in advanced PAH (Fig 1F). These findings suggest a paradoxical interplay between proliferative and anti-inflammatory transcriptional programs in PAH pathogenesis.

### 3.2. PPI network identifies CD44 as a central hub gene in PAH-associated inflammation

To prioritize inflammation-related candidates, the 670 upregulated genes were intersected with the GeneCards "Inflammation" gene set (score>5), yielding 47 overlapping genes (Fig 2A). A PPI network constructed using STRING (confidence score>0.7) revealed dense connectivity among these candidates (Fig 2B). CD44 consistently ranked as the top-scoring hub gene across multiple topological algorithms, including Betweenness, Bottleneck, Closeness, EPC (Edge Percolated Component), Radiality, Degree, MNC (Maximum Neighborhood Component), EcCentricity, and Stress. (Fig 2C–J). CD44's centrality in the network underscores its potential role in coordinating inflammatory signaling. In this dataset, the expression level of CD44 was also elevated in the IPAH group (Fig 2K). Gene Set Enrichment Analysis (GSEA) of CD44-associated genes further linked its activity to "leukocyte migration" (NES=2.4, p=0.002) and "extracellular matrix organization" (NES=2.1, p=0.008), supporting its involvement in vascular inflammation and remodeling.

To further validate the PPI network results, we performed a Spearman correlation-based co-expression analysis using the bulk RNA-seq dataset (S2 Fig). We identified the top 30 genes most strongly co-expressed with CD44, among which EFHD2 and MGLL overlapped with the inflammation-related upregulated genes in the PPI network.

### 3.3. Single-cell RNA-seq analysis identifies CD44-expressing cells as key contributors to PAH pathogenesis

To investigate the expression landscape of CD44 at single-cell resolution, we performed scRNA-seq analysis on lung tissues from 3 PAH patients and 3 controls (GSE228644). Quality control metrics, including gene count (nFeature_RNA), UMI count (nCount_RNA), and mitochondrial gene percentage (percent.mt), were comparable across groups (Fig 3A, B). A total of 2,000 highly variable genes (HVGs) were identified for downstream analysis (Fig 3C). Dimensionality reduction and clustering revealed 24 distinct cell clusters, annotated into major cell types such as fibroblasts, SMC, endothelial cells, epithelial cells, T cells, macrophages, and others (Figs 3D and S3). Feature plots showed that CD44 was widely expressed, with particularly high levels in fibroblasts, immune cells and SMCs (Figs 3E, F and S4A, B), consistent with the expression pattern in the Human Protein Atlas (HPA) database (S4C Fig). Importantly, CD44 expression was significantly upregulated in IPAH lungs compared to controls, both across all cells (p<2.2e-16, Wilcoxon test; Fig 3G) and specifically in SMCs (p=2.6e-13, Wilcoxon test; Fig 3H), supporting its potential role in disease pathogenesis.

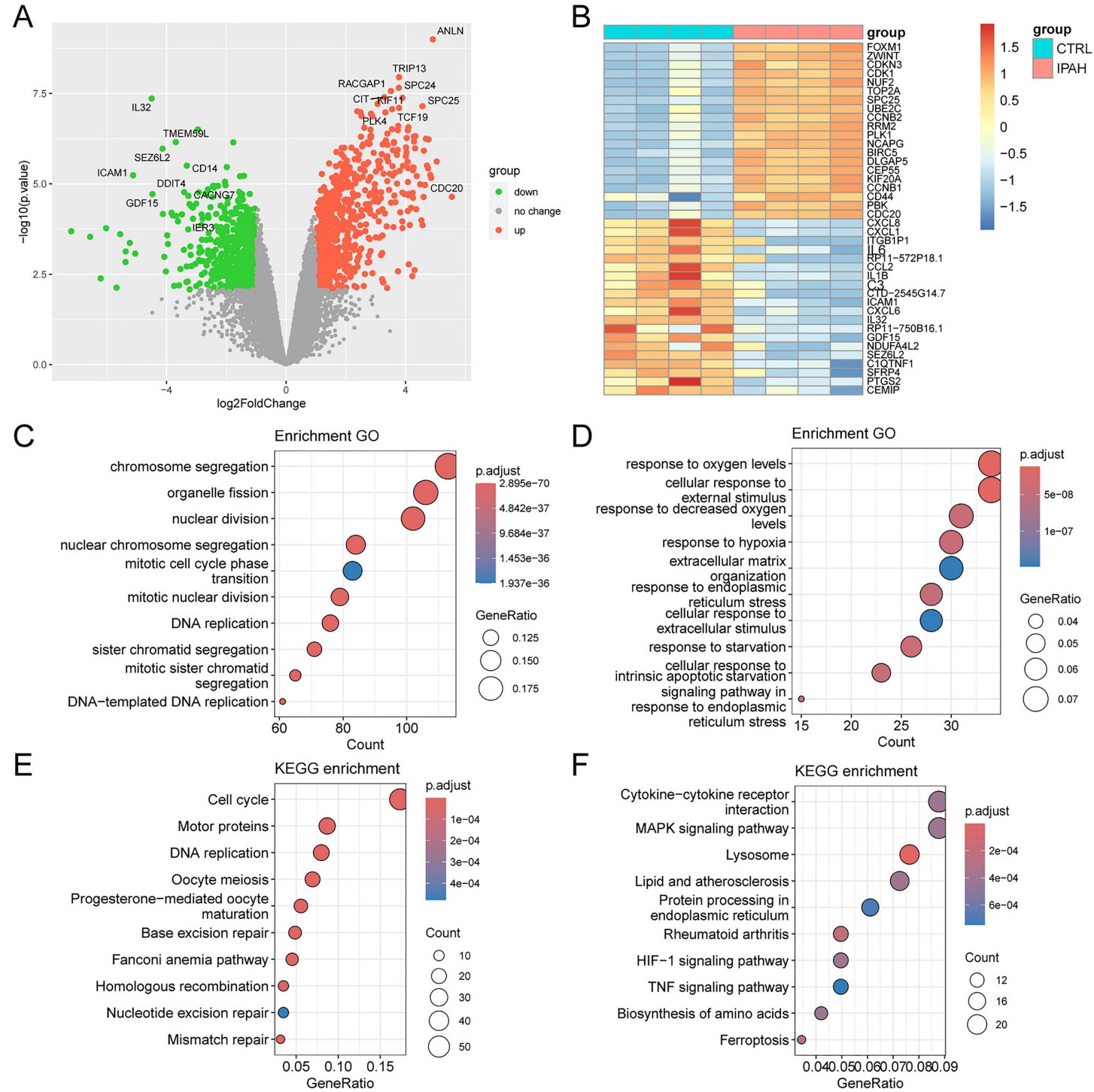

**Fig 1. Identification of PAH-associated genes based on bulk RNA-seq analysis.** A: Volcano plot showing DEGs between PAH patients and controls. B: Heatmap displaying the expression of 20 upregulated and 20 downregulated genes from the DEGs. C-D: GO enrichment analysis for the upregulated genes (C) and downregulated genes (D), respectively. E-F: KEGG pathway analysis for the upregulated genes (E) and downregulated genes (F), respectively.

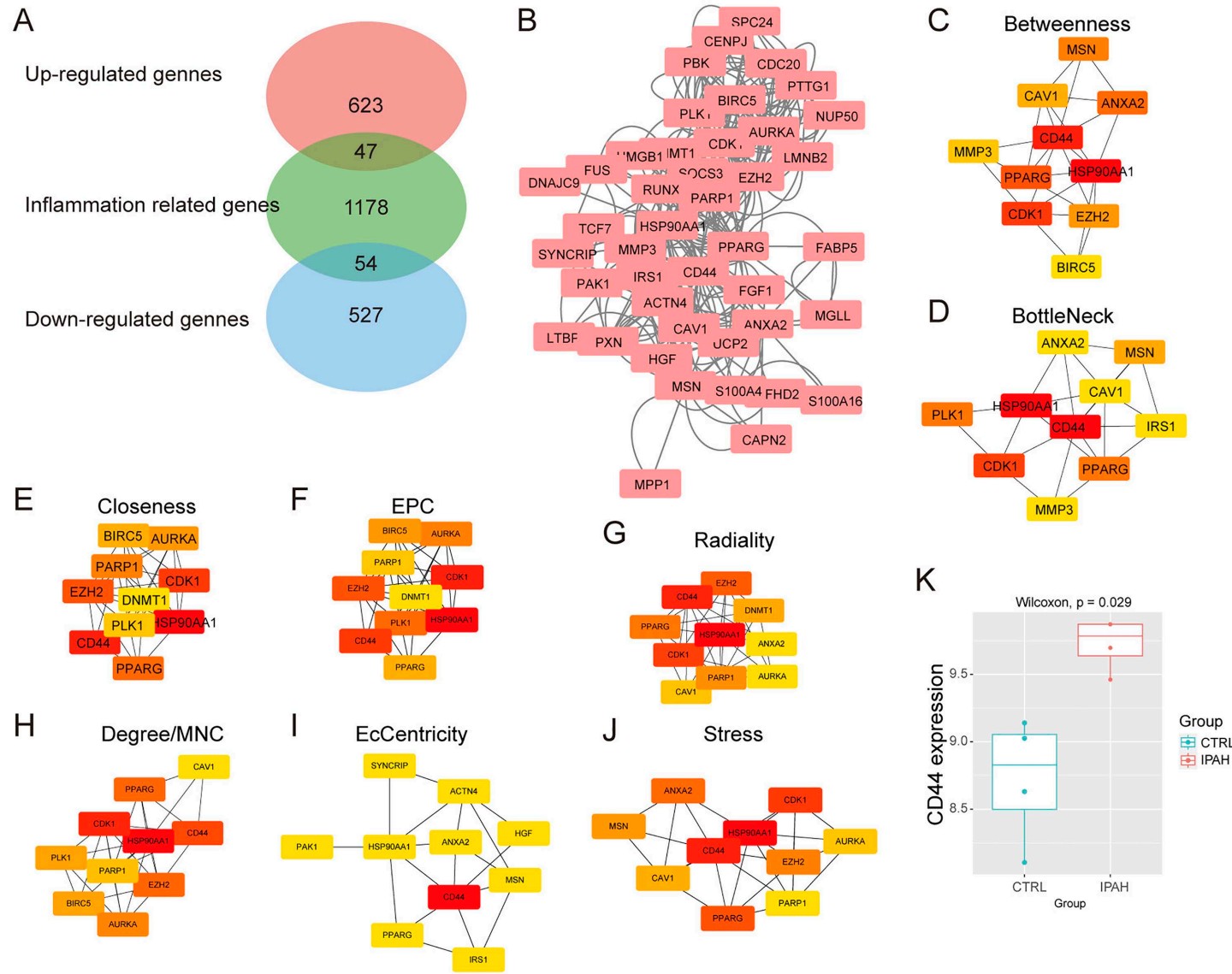

**Fig 2. Identification of core inflammation-related genes in IPAH.** A: Venn diagram showing the overlap of inflammation-related DEGs. B: PPI network of inflammation-associated upregulated DEGs. C-J: Sub-topological analysis of the top 10 hub genes identified using different ranking algorithms in the cytoHubba plugin, including Betweenness, Bottleneck, Closeness, EPC, Radiality, Degree, MNC, EcCentricity, and Stress. Each panel displays the top 10 genes ranked by the respective algorithm. K: Comparison of CD44 expression between PAH patients and controls in bulk RNA-seq data (Wilcoxon test, p = 0.029).

## 3.4. Molecular docking and dynamics simulations reveal high-affinity interaction between progesterone and CD44

To explore CD44-targeted therapeutic strategies, molecular docking was performed between the CD44 HA-binding domain (PDB: 1UUH) and progesterone (CID: 5994). The docking model suggested that progesterone binds to the CD44 protein with high affinity, locating in a surface groove region (Fig 4A, B). This revealed that progesterone binds to CD44 with a higher predicted affinity (−7.2 kcal/mol) than HA (−5.1 kcal/mol), the natural ligand of CD44 (Figs 4B–D and S5).

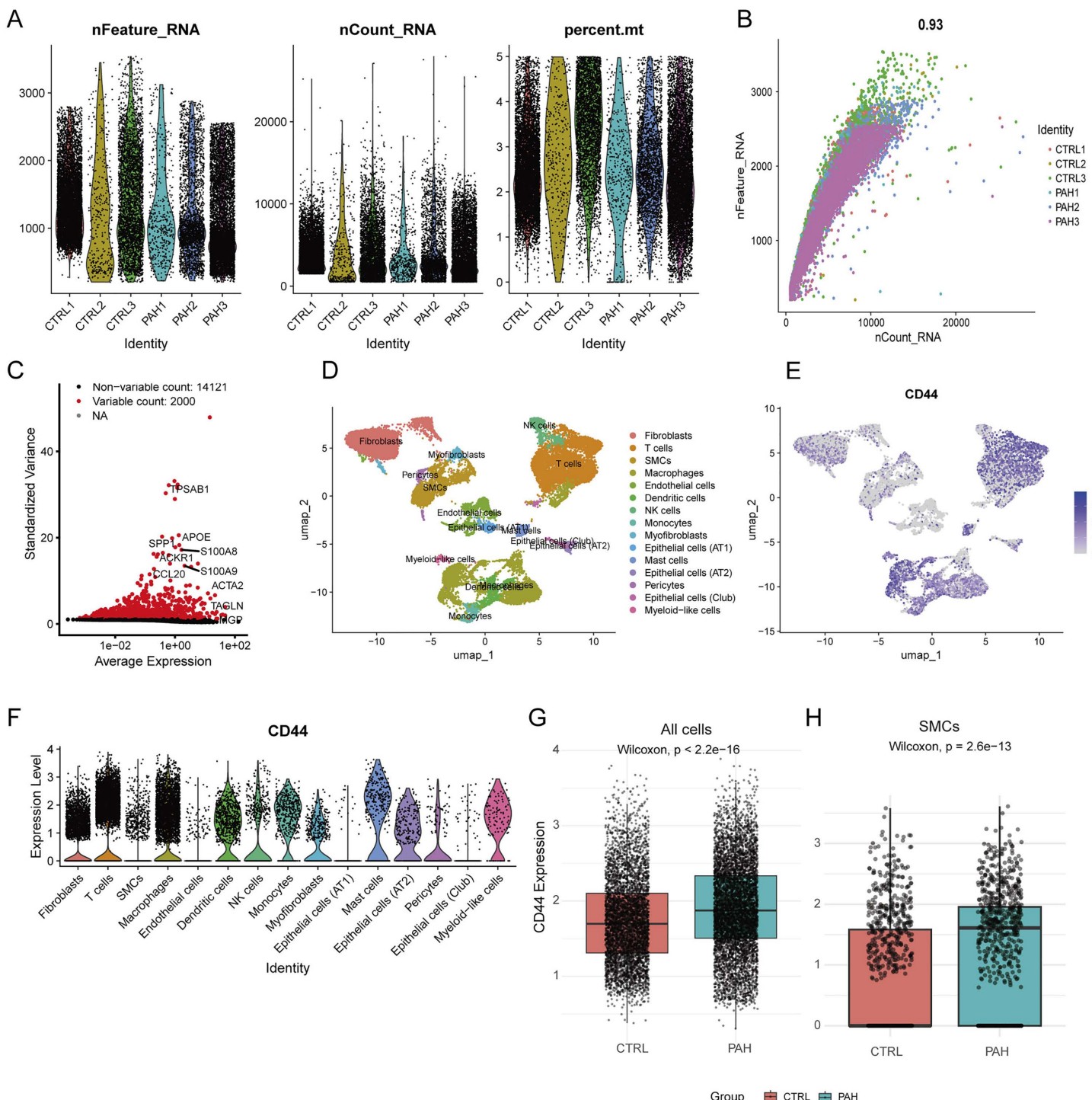

**Fig 3. scRNA sequencing analysis of GSE228644 in IPAH and healthy groups.** A: Summary of the features, counts, and mitochondrial gene percentage for each sample. B: Correlation between gene expression levels and counts in each sample. C: HVGs are highlighted in red, with the top 10 HVGs labeled. D: UMAP visualization identified 15 major lung cell populations across all samples. E-F: UMAP feature plot and Violin plot show that CD44 is broadly expressed across multiple cell types. G-H: Expression of CD44 in total cells (G, Wilcoxon test, p < 2.2e-16) and SMCs (H, Wilcoxon test, p = 2.6e-13) in IPAH and healthy groups.

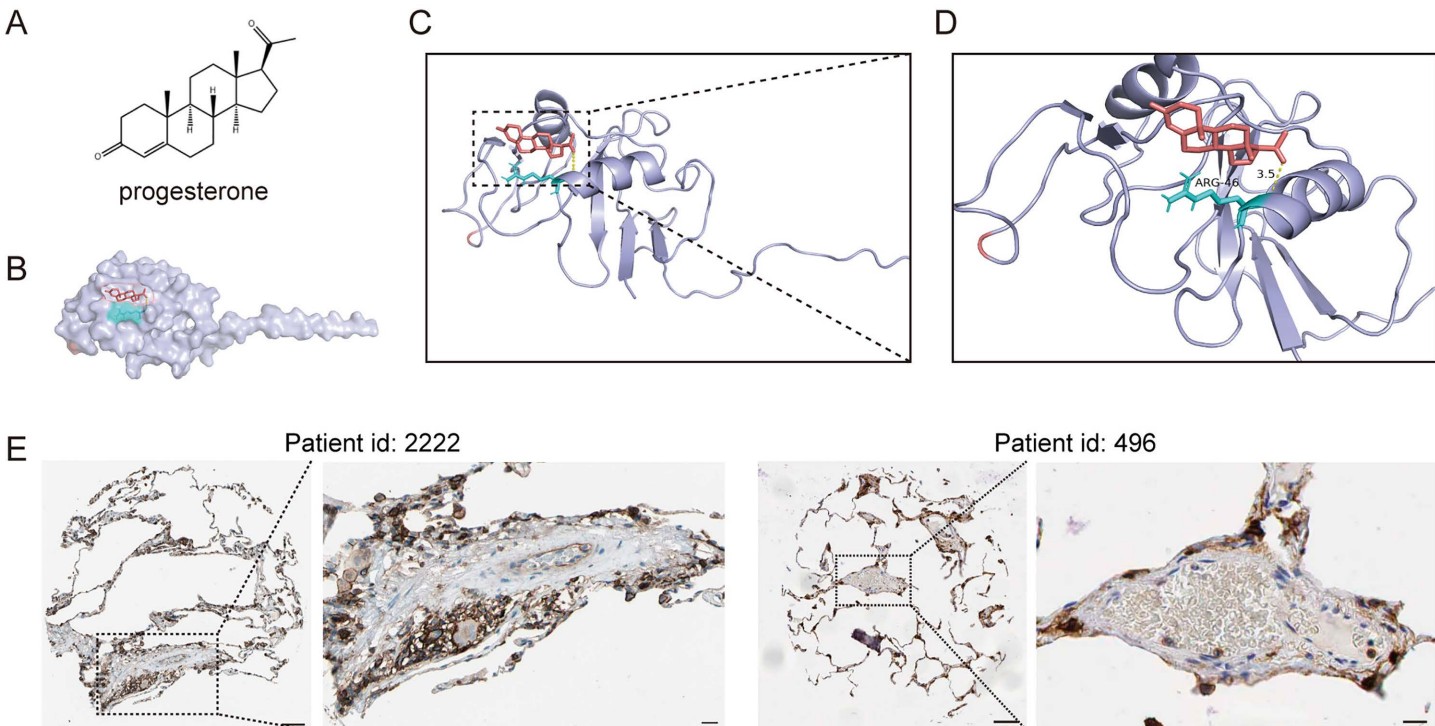

**Fig 4. Progesterone interacts with CD44 and CD44 is highly expressed in the pulmonary vasculature of IPAH patients.** A: Chemical structure of progesterone. B: Molecular docking surface model showing progesterone (red) binding to the CD44 protein. C: Docking pose of progesterone binding to CD44. D: Hydrogen bond interactions between progesterone and CD44 Arg46 (cyan). E: Immunohistochemical image of CD44 in lung tissue sections from two IPAH patients (ID: 2222 and 496) in the HPA database. Scale bars: 100 µm (overview), 50 µm (magnified insets).

Progesterone formed hydrogen bonds with Arg46 (2.2 Å) within the HA-binding pocket, mimicking HA interactions (Fig 4C, D). These results position progesterone may function as a potential CD44 antagonist.

Furthermore, CD44 protein expression was confirmed by immunohistochemistry using data from the HPA database. Lung tissue sections from IPAH patients showed strong CD44 staining in pulmonary vascular walls, especially in the smooth muscle and endothelial layers. Representative images from patient 2222 and patient 496 illustrated enhanced CD44 expression in vascular lesions (Fig 4E).

## 4. Discussion

PAH is characterized by progressive vascular remodeling, in which SMC proliferation and inflammation play central roles [21]. This study establishes CD44 as a central regulator of inflammation in PAH, bridging transcriptional dysregulation, inflammation, and vascular pathology. Through integrative multi-omics analysis, we identified CD44 as a central upregulated gene in PAH, showing consistent overexpression across bulk and single-cell transcriptomic profiles, especially in SMCs. Inflammatory cytokines drive PAH pathogenesis by promoting endothelial-SMCs activation and vascular remodeling [9]. The observed CD44 overexpression in SMCs underscores its role in sustaining inflammatory loops, and this inflammation may trigger vascular remodeling and exacerbate PH.

CD44 is a multifunctional cell adhesion molecule that binds extracellular matrix components to mediate cell adhesion, migration, and inflammatory responses. Accumulating evidence supports a direct role of CD44 in SMC activation and inflammatory response. CD44-v6 is a marker of synthetic SMCs during active vascular remodeling in ductus arteriosus

closure [22]. CD44 promotes atherosclerosis by mediating inflammatory cell recruitment, vascular SMC activation, and phenotypic switching to a pro-inflammatory synthetic state [18]. CD44 and other hyaluronan-binding molecules are ubiquitously expressed across multiple cell types [23]. Hyaluronan directly binding CD44 could promote SMC migration and proliferation [24,25]. Conversely, high-molecular-weight hyaluronan exerts inhibitory effects on SMC proliferation [18,26]. These findings are consistent with our observed CD44 upregulation in SMC clusters, further supporting its pathogenic role in PAH vascular remodeling and highlighting its potential as a therapeutic target.

The molecular docking studies provide a novel insight into the interaction between progesterone and CD44, which may offer therapeutic potential. Progesterone's ability to modulate CD44 signaling could represent a promising strategy for reducing inflammation and improving clinical outcomes in PAH patients. Progesterone holds significant therapeutic promise as an anti-inflammatory drug and immunomodulator [27]. Notably, progesterone induces human leukocyte antigen-g expression in vascular endothelial and SMC to protect against allograft rejection and vasculopathy [28]. In clinical, female PAH patients exhibit superior right ventricle function and a better prognosis than their male counterparts [29], suggesting involvement of sex hormones in PAH pathobiology. Additionally, progesterone receptors are expressed in pulmonary SMC, suggesting additional vasodilatory mechanisms [30]. Our molecular docking analysis further highlights progesterone as a potential ligand for CD44, offering a mechanistic basis for modulating its activity in the vascular microenvironment.

However, systemic CD44 inhibition poses significant risks due to its broad expression across multiple immune and non-immune cell types [23]. Blanket inhibition of CD44 may therefore impair normal immune surveillance, increase susceptibility to infections, or interfere with wound healing and tissue regeneration. Hence, systemic CD44 inhibition risks impairing physiological immune surveillance, necessitating tissue-specific delivery strategies.

Despite the robust multi-omics integration, our study has several limitations. The small sample sizes in both bulk (n = 4) and single-cell RNA-seq (n = 3) analyses may limit the statistical power and generalizability of the findings, though they still offer valuable insights into CD44 expression in PAH. The proposed interaction between progesterone and CD44 is computationally predicted and lacks experimental validation, which remains a limitation of this study. Additionally, the lack of large-scale clinical validation highlights the need for further studies in broader patient populations to confirm the role of CD44 in PAH. Therefore, future studies involving larger, independent patient cohorts are necessary to validate and extend these results.

Together, our findings highlight the importance of integrating multi-omics approaches, such as bulk RNA-seq and single-cell RNA-seq, in understanding the complex molecular mechanisms underlying PAH. By promoting SMC inflammation and interacting with immune components, CD44 may serve as both a biomarker and a therapeutic target for halting or reversing vascular remodeling in PAH. Future studies should also focus on elucidating the role of CD44 in other immune cell types and its potential interactions with other signaling pathways involved in PAH.

## 5. Conclusion

This study identifies CD44 as a critical gene in the pathogenesis of PAH with multi-omics approach integrating bulk RNA sequencing, single-cell transcriptomics, and molecular docking. Our findings establish CD44 as a critical mediator of inflammation and vascular remodeling in PAH, and a promising target for therapeutic intervention. Future studies focusing on the regulatory mechanisms of CD44 in SMCs and its modulation by endogenous or exogenous ligands may open new avenues for PAH treatment.

## Supporting information

**S1 Fig. Quality control and sample correlation of bulk RNA sequencing data.** A: Boxplots showing gene expression distributions across all samples. B: Heatmap of Pearson correlation coefficients between all samples. C: Hierarchical clustering of samples based on transcriptomic similarity, with CTRL samples in blue and IPAH samples in red.
(TIF)

**S2 Fig. Identification of genes co-expressed with CD44 in PAH lung tissues.** A: Heatmap showing expression patterns of the top 30 genes most strongly co-expressed with CD44 based on Spearman correlation in bulk RNA-seq data from IPAH and control samples. B: Bar plot ranking the top 30 genes by absolute Spearman correlation coefficient.
(TIF)

**S3 Fig. Cell clustering and marker gene expression in lung scRNA-seq data.** A: UMAP plot showing integration of cells from CTRL and IPAH groups, colored by condition. B: UMAP plot displaying 24 identified clusters, each labeled with its cluster number. C: Heatmap of representative marker genes across clusters.
(TIF)

**S4 Fig. Expression of CD44 in various cell types and lung tissues.** A: Schematic representation of CD44 expression across annotated cell types. B: Dot plot analysis of CD44 expression across distinct cell types in this lung scRNA-seq data. C: Validation of CD44 protein expression across distinct cell types in human lung tissue from the HPA database.
(TIF)

**S5 Fig. Molecular docking of progesterone with the CD44 protein.** A: Docking pose of HA binding to CD44. B: Hydrogen bond interactions between HA and key residues of CD44, including ARG-46, SER-109, and GLN-113. C Predicted binding affinities of progesterone and the natural ligand HA with CD44.
(TIF)

**S1 Table. Complete list of DEGs from the bulk RNA-seq analysis, including gene symbols, log2 fold changes, p-values, and adjusted p-values.**
(XLSX)

**S2 Table. Full results of GO term and KEGG pathway enrichment analyses, including terms, gene counts, and FDR values.**
(XLSX)

**S3 Table. Marker gene statistics and cell type annotations for all single-cell clusters, including p-values, fold-changes, expression frequencies, and curated cell identities.**
(XLSX)

## Author contributions

**Conceptualization:** Haiyan Qi.

**Data curation:** Wei Chen.

**Formal analysis:** Wei Chen, Lingling Zhang.

**Funding acquisition:** Haiyan Qi.

**Investigation:** Wei Chen, Lingling Zhang, Haiyan Qi.

**Methodology:** Wei Chen, Lingling Zhang, Haiyan Qi.

**Resources:** Wei Chen.

**Software:** Wei Chen.

**Validation:** Haiyan Qi.

**Visualization:** Haiyan Qi.

**Writing – original draft:** Haiyan Qi.

**Writing – review & editing:** Haiyan Qi.

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
