## [Decision Letter · Decision Letter 0]

23 Jun 2025

Dear Dr. Qi,

Thank you for submitting your manuscript to PLOS ONE. After careful consideration, we feel that it has merit but does not fully meet PLOS ONE’s publication criteria as it currently stands. Therefore, we invite you to submit a revised version of the manuscript that addresses the points raised during the review process.

We look forward to receiving your revised manuscript.

Kind regards,

Naseem Ahamad

Academic Editor

PLOS ONE

Journal Requirements:

“This work was supported by grants from Scientific Research Foundation of the Education Department of Jiangxi Province, China (No. GJJ2200446 to H.Q.) and Jiangxi Provincial Graduate Innovation Special Foundation (No. YC2024-B178 to H.Q. and No. YC2023-B166 to W.C.)”

“This work was supported by grants from Scientific Research Foundation of

the Education Department of Jiangxi Province, China (No. GJJ2200446 to H.Q.) and Jiangxi Provincial Graduate Innovation Special Foundation (No. YC2024-B178 to H.Q. and No. YC2023-B166 to W.C.).”

“This work was supported by grants from Scientific Research Foundation of

the Education Department of Jiangxi Province, China (No. GJJ2200446 to H.Q.) and Jiangxi Provincial Graduate Innovation Special Foundation (No. YC2024-B178 to H.Q. and No. YC2023-B166 to W.C.).”

Additional Editor Comments (if provided):

Thank you for your submission. The reviewers have given important feedback that needs to be addressed before your work can be considered for publication.

Please revise your manuscript to address their concerns and include a detailed response letter explaining how you have addressed each point raised by the reviewers. Their enthusiasm for your molecular docking approach indicates that with these revisions, your manuscript has strong potential for publication.

Reviewers' comments:

Reviewer's Responses to Questions

**Comments to the Author**

1. Is the manuscript technically sound, and do the data support the conclusions?

Reviewer #1: Yes

Reviewer #2: Yes

2. Has the statistical analysis been performed appropriately and rigorously?

Reviewer #1: No

Reviewer #2: Yes

3. Have the authors made all data underlying the findings in their manuscript fully available?

Reviewer #1: No

Reviewer #2: Yes

4. Is the manuscript presented in an intelligible fashion and written in standard English?

Reviewer #1: No

Reviewer #2: Yes

Reviewer #1: 1. No correction for batch effects or potential confounders. No ROC analysis or machine learning models to predict CD44’s clinical significance.

2. Gene expression and KEGG/GO data need to added in supplementary data.

Reviewer #2: The way the authors represent their work is fabulous and I found all the work are prefect for the molecular docking level. based on these findings future research should focus on understanding CD44's regulatory mechanisms

and developing targeted therapies in the wet lab.

**Do you want your identity to be public for this peer review?** For information about this choice, including consent withdrawal, please see our Privacy Policy

Reviewer #1: **Yes: ** Prince Kumar

Reviewer #2: **Yes: ** Kannan Kanthaiah

---

## [Author Response · Author response to Decision Letter 1]

21 Aug 2025

Response to Reviewer Comments

We sincerely thank the reviewers for their thoughtful and constructive feedback. We have carefully revised the manuscript according to the comments and provide detailed responses below. All modifications have been marked in the revised manuscript.

1. “The introduction lacks a critical discussion on prior CD44 studies in PAH (Ref 15 could be elaborated).”

Response:

Thank you for pointing this out. We have revised the Introduction section to include a more detailed discussion on previous studies involving CD44 in the context of PAH, particularly elaborating on Ref 15. This provides a more comprehensive background and better highlights the novelty of our study.

2. “The bulk RNA-seq analysis was conducted on samples from only 4 iPAH patients and 4 healthy controls. Similarly, the single-cell RNA-seq analysis used data from 3 iPAH patients and 3 controls. These are relatively small sample sizes, which can limit the statistical power of the study and the generalizability of the findings to the broader PAH patient population. This should acknowledged as a limitation.”

Response:

We agree that the small sample sizes are a limitation. We have added a clear statement acknowledging this in the Discussion section, emphasizing that our findings require validation in larger and independent clinical cohorts.

3. “Add supplementary files with complete DEG lists and enriched pathway tables.”

Response:

Done. We have now included the complete list of differentially expressed genes and enriched pathways (GO and KEGG) in Supplementary Tables S1 and S2, respectively.

• Supplementary Table S1: Complete list of differentially expressed genes (DEGs) from the bulk RNA-seq analysis, including gene symbols, log2 fold changes, p-values, and adjusted p-values.

• Supplementary Table S2: Full results of Gene Ontology (GO) and KEGG pathway enrichment analyses, including terms, gene counts, and FDR values.

4. “The Materials and Methods section and the Data Availability statement consistently refer to the single-cell RNA-seq dataset as GSE228644. However, the legend for Figure 3, which presents the scRNA-seq results, states the data is from GSE231993. This is a clear inconsistency that needs to be corrected.”

Response:

Thank you for noticing this. We have corrected the inconsistency. The correct dataset used for scRNA-seq is GSE228644, and this has been updated throughout the manuscript, including the Figure 3 legend.

5. “Enrichment tools and thresholds are standard, but exact GO/KEGG terms and p-values should be added in at least in Supplementary Data.”

Response:

We thank the reviewer for the comment. We used standard enrichment analysis parameters, specifically setting pvalueCutoff = 0.05 and qvalueCutoff = 0.2 to identify significantly enriched GO and KEGG terms. The full enrichment results, including term names and statistical values, have been provided in the Supplementary Data.

Supplementary Table S2: Full results of Gene Ontology (GO) and KEGG pathway enrichment analyses, including terms, gene counts, and FDR values.

6. “PPI networks predict interactions but do not confirm functional effects experimentally. Lacks complementary co-expression analysis (from RNA-seq data) to strengthen findings.”

Response:

We appreciate the reviewer’s insightful suggestion regarding the integration of co-expression analysis to complement the PPI network findings. In response, we performed a Spearman correlation-based co-expression analysis using the bulk RNA-seq dataset. Among the top 30 genes most strongly co-expressed with CD44, we identified EFHD2 and MGLL, both of which also appeared in the PPI network constructed from inflammation-related upregulated DEGs. This overlap strengthens the functional relevance of these genes within the CD44-associated inflammatory module and supports the biological significance of our PPI network predictions. We have added the co-expression analysis results to the revised manuscript (Figure X).

7. “The molecular docking results suggesting a high-affinity interaction between progesterone and CD44 are based on computational predictions. While this provides a novel hypothesis for therapeutic intervention, these findings are preliminary and require experimental validation (e.g., in vitro binding assays, cellular or animal model studies) to confirm the interaction and its functional significance in a biological context.”

Response:

We agree. We have added a paragraph in the Discussion section clearly stating that the molecular docking results are computational predictions and must be validated experimentally through functional studies such as in vitro binding assays and animal models.

8. “Molecular docking analysis seems overexerted as the docking energy (−7.2 kcal/mol) is

reasonable but not definitive. Moreover, lacks comparative analysis with known CD44 ligands (e.g., HA analogs) and no molecular dynamics simulation was performed to support binding stability. Therefore looking at aforesaid limitations, the identification of progesterone as a CD44 ligand is speculative, thus needs a more cautious tone or follow-up validation. Statements like “These results position progesterone as a potential CD44 antagonist” are premature without functional assays.”

Response:

Thank you for the insightful comment. We agree that molecular docking alone provides preliminary evidence and requires experimental validation. Accordingly, we adjust our statements. Furthermore, we performed a comparative docking analysis using hyaluronic acid (HA), a well-established CD44 ligand, and showed that progesterone exhibited a higher predicted binding affinity (−7.2 kcal/mol vs. −5.1 kcal/mol). These results are now included in the revised Figure S5 and discussed appropriately. We also acknowledged the absence of MD simulations as a limitation, and emphasized the need for further validation through in vitro binding assays and functional studies.

9. “Statement: “highlighting systemic immune suppression in advanced PAH” contradicts the theme that inflammation is increased and requires further clarification.”

Response:

We appreciate this insightful comment. We agree that the original statement “highlighting systemic immune suppression in advanced PAH” may seem inconsistent with the broader theme of inflammation. To address this, we have revised the sentence to clarify that the observed downregulation of certain inflammatory pathways reflects immune dysregulation or exhaustion, which is commonly seen in chronic inflammatory diseases like advanced PAH.

10. “Author should discuss about unintended consequences of CD44 inhibition.”

Response:

This is a valuable point. We have added a paragraph in the Discussion addressing potential unintended consequences of CD44 inhibition, such as impaired tissue repair or altered immune surveillance, which should be considered in therapeutic development.

Minor Comments

• “Fix typographical and grammatical errors throughout.”

Response:

We have carefully proofread the manuscript and corrected all identified typographical and grammatical errors.

• “Improve figure quality and annotations.”

Response:

We have enhanced the resolution of all figures and improved labeling and annotation to ensure clarity and readability.

---

## [Editor Report · Decision Letter 1]

5 Sep 2025

CD44 as a Novel Therapeutic Target in Pulmonary Arterial Hypertension: Insights from Multi-Omics Integration and Molecular Docking

PONE-D-25-09501R1

Dear Dr. Qi,

We’re pleased to inform you that your manuscript has been judged scientifically suitable for publication and will be formally accepted for publication once it meets all outstanding technical requirements.

Kind regards,

Naseem Ahamad

Academic Editor

PLOS ONE
---

## [Editor Report · Acceptance letter]

PONE-D-25-09501R1

PLOS ONE

Dear Dr. Qi,

I'm pleased to inform you that your manuscript has been deemed suitable for publication in PLOS ONE. Congratulations! Your manuscript is now being handed over to our production team.

Kind regards,

on behalf of

Dr. Naseem Ahamad

Academic Editor

PLOS ONE